# Steady Enhancement in Photovoltaic Properties of Fluorine Functionalized Quinoxaline-Based Narrow Bandgap Polymer

**DOI:** 10.3390/molecules24010054

**Published:** 2018-12-24

**Authors:** Zhonglian Wu, Huanxiang Jiang, Xingzhu Wang, Lei Yan, Wei Zeng, Xiu-Gang Wu, Haiyu Zhuang, Wen Zhu, Renqiang Yang

**Affiliations:** 1School of Materials Engineering, Jiangsu University of Technology, Changzhou 213001, China; wuzhonglian@jsut.edu.cn (Z.W.); zhuanghaiyu@jstu.edu.cn (H.Z.); czzwen@163.com (W.Z.); 2Qingdao Institute of Bioenergy and Bioprocess Technology, Chinese Academy of Sciences, Qingdao 266101, China; jianghx@qibebt.ac.cn; 3College of Chemistry, Xiangtan University, Xiangtan 411105, China; yanlei@xtu.edu.cn (L.Y.); jonewel@gmail.com (W.Z.); 4Southern University of Science and Technology, Shenzhen 518055, China; 5Jiangsu Key Laboratory of Environmentally Friendly Polymeric Materials, School of Materials Science & Engineering, Changzhou University, Changzhou 213164, China; xgwu16@126.com

**Keywords:** polymer solar cells, bulk heterojunction, quinoxaline, narrow bandgap conjugated polymer, synthesis

## Abstract

To investigate the influence of fluoride phenyl side-chains onto a quinoxaline (Qx) unit on the photovoltaic performance of the narrow bandgap (NBG) photovoltaic polymers, herein, two novel NBG copolymers, **PBDTT-DTQx** and **PBDTT-DTmFQx**, were synthesized and characterized. 2-ethylhexylthiothiophene-substituted benzodithiophene (BDTT), 2,3-diphenylquinoxaline (DQx) [or 2,3-bis(3-fluorophenyl)quinoxaline (DmFQx)] and 2-ethylhexylthiophene (T) were used as the electron donor (D) unit, electron-withdrawing acceptor (A) unit and π-bridge, respectively. Compared to non-fluorine substituted **PBDTT-DTQx**, fluoride **PBDTT-DTmFQx** exhibited a wide UV-Vis absorption spectrum and high hole mobility. An enhanced short-circuit current (*J*_sc_) and fill factor (FF) simultaneously gave rise to favorable efficiencies in the polymer/PC_71_BM-based polymer solar cells (PSCs). Under the illumination of AM 1.5G (100 mW cm^−2^), a maximum power conversion efficiency (PCE) of 6.40% was achieved with an open-circuit voltage (*V*_oc_) of 0.87 V, a *J*_sc_ of 12.0 mA cm^−2^ and a FF of 61.45% in **PBDTT-DTmFQx**/PC_71_BM-based PSCs, while **PBDTT-DTQx**-based devices also exhibited a PCE of 5.43%. The excellent results obtained demonstrate that **PBDTT-DTmFQx** by fluorine atom engineering could be a promising candidate for organic photovoltaics.

## 1. Introduction

Polymer solar cells (PSCs) have been attracting tremendous interest as next-generation solar cells due to their unique advantages of low cost fabrication, light weight and potential for a large area through solution-processing [1,2,3,4,5,6,7]. At present, the state-of-the-art structure of PSCs is based on the bulk heterojunction (BHJ) type, where the photoactive layer is composed of a blend of a conjugated polymer as the donor material and a fullerene derivative or non-fullerene fuse-ring molecule as the acceptor material [8,9,10]. Through synergistic work of the development of efficient conjugated polymers and device optimization, a series of PSCs whose power conversion efficiency (PCE) is over 10% have been obtained [4,11,12,13,14]. However, the relationship between molecular structures and good photovoltaic performance is still unclear. Hence, to obtain high-efficient PSCs, the donor materials are required to have narrow bandgap energy and suitable molecular energy levels to maximize the short-circuit current (*J*_sc_), open-circuit voltage (*V*_oc_) and high carrier mobility to facilitate carrier transport. These goals can be achieved by tuning the molecular structures of donor–acceptor (D–A) conjugated polymers. Of these, quite a few donor and acceptor units have been developed in the past decade, such as quinoxaline (Qx) [15,16] or 2,5-dihydropyrrolo[3,4-c]-pyrrole-1,4-dione (DPP) [17] as the acceptor unit and 2,7-fluorene [18], 2,7-carbazole [19], benzo[1,2-b:4,5-b′]dithiophene (BDT) as the donor unit [20,21]. Among these various donor units, the BDT unit has become one of the most successful frameworks in the design and synthesis of efficient donor materials since it was first used for PSCs in 2009, owing to its rigid and planar molecular structure [20]. Two-dimensional conjugated BDT units were developed for the synthesis of donor materials, which not only broaden the absorption spectra, but also enhance the hole mobilities [22,23]. For example, the Hou group [11] reported a high power conversion efficiency (PCE) of 14.2% with an outstanding FF of 0.76 based on a DTBDT unit.

As a strong electron acceptor unit, quinoxaline (Qx) is widely used as the skeleton of a D–A type conjugated polymer for PSC applications [24,25,26,27,28,29,30,31]. The molecular structure of the Qx unit can be easily modified at the 2,3-position and the 6,7-position to tune the optoelectronic properties. As is well known, the fluorine atom has a very small atomic radius (only slightly larger than that of the H atom), strong electronegativity and intermolecular weak non-covalent interactions (such as F∙∙∙H, F∙∙∙S, F∙∙∙π, etc.). Recently, introducing fluorine atoms in conjugated polymers as the donor materials in PSCs was generally expected to modulate the optical and electrochemical properties, and molecular packing to facilitate carrier transport to enhance photovoltaic properties [32]. The introduction of a fluorine atom into the Qx unit of low bandgap polymers has been regarded as one effective strategy for improving the photovoltaic properties of PSCs [25,29,30,31]. For example, the NBG polymer with a fluoride Qx unit at the 5,6-position as the donor, and PC_71_BM as the acceptor, achieved a high PCE value of 8.0% [29]. The Liu group synthesized the polymer based on a Qx unit with a 4-fluorophenyl side chain at the 2,3-position. Its optimized PSCs based on the polymer/PC_71_BM demonstrated an outstanding PCE value of 7.2% with a *V*_oc_ of 0.87 V, *J*_sc_ of 11.4 mA cm^−2^, and a fill factor (FF) of up to 73% [25].

Encouraged by these results, in this study we report the synthesis and properties of non-fluorinated and fluorinated Qx units based on narrow-bandgap (NBG) polymers, namely **PBDTT-DTQx** and **PBDTT-DTmFQx** (Scheme 1), in which a strong donor unit of 2-ethylhexylthiothiophene-substituted benzodithiophene (BDTT) has been introduced in the main chains. We expected that the grafting of fluorine atoms can further pare away the highest occupied molecular orbital (HOMO) energy level and improve the charge carrier mobility of the corresponding polymer, and likewise, also enhance the photovoltaic properties for its polymer in PSCs. As a result, **PBDTT-DTmFQx**/PC_71_BM-based devices presented a maximum PCE of 6.40%, and were obtained with a *V*_oc_ of 0.87 V, *J*_sc_ of 12.0 mA cm^−2^ and FF of 61.45%, while non-fluorinated **PBDTT-DTQx**-based devices demonstrated a PCE of 5.43%. This result demonstrates that the fluorine-substituted side-chain on the acceptor unit is a useful strategy to fine tune energy levels of NBG polymers.

## 2. Results and Discussion

### 2.1. Synthesis and Thermal Stability

The synthetic routes of the monomers and the copolymers **PBDTT-DTQx** and **PBDTT-DTmFQx** are depicted in Scheme 1. The compounds **2a**–**b** were prepared by a reduction reaction using NaBH_4_ with compound 1 and then a condensation reaction using sodium acetate as a catalyst with benzil or 3,3′-difluorobenzil in good yield. The compounds **3a**–**b** were obtained between compound **2a**–**b** and tributyl(3-(2-ethylhexyl)-thiophen-2-yl)stannane in excellent yield, by a Stille coupling reaction using Pd(PPh_3_)_4_ as the catalyst,. The compounds **3a**–**b** were brominated with *N*-bromosuccinimide (NBS) to afford the monomers **4a**–**b**. Through a typical Stille coupling polymerization, **PBDTT-DTQx** and **PBDTT-DTmFQx** were obtained between **4a**–**b** and **5** with tris(dibenzylideneacetone) dipalladium [Pd_2_(dba)_3_] and tri(*o*-tolyl)phosphine [P(*o*-tol)_3_] as the catalysts in toluene at 110 °C. The copolymers were carefully purified by continuous Soxhlet extraction with *n*-hexane, acetone, and chloroform successively. The chloroform fraction was concentrated under reduced pressure, and then precipitated in methanol and collected by filtration. The chemical structure of all compounds was confirmed by ^1^H-NMR and ^13^C-NMR. Both of the resulting copolymers were characterized by elemental analysis. High-quality ^1^H-NMR spectra for the copolymers could not be obtained due to their limited solubility in CDCl_3_ at room temperature (RT). The molecular weight of the copolymers was determined by gel permeation chromatography (GPC) using tetrahydrofuran as the eluent and polystyrene as the standard (Table 1). The average molecular weights (*M*_n_) of **PBDTT-DTQx** and **PBDTT-DTmFQx** were 18.5 kDa (PDI = 2.4) and 21.2 kDa (PDI = 2.5), respectively. As four 2-ethylhexyl side chains on repeating units of the copolymers were appended on the backbone, both the copolymers showed good solubility, and could be easily dissolved in the commonly used solvents for device fabrication such as THF, toluene, xylene, chlorobenzene (CB) and *o*-dichlorobenzene (ODCB). Thermal stability of the copolymers was surveyed by thermogravimetric analysis (TGA). As shown in Figure 1 and Table 1, **PBDTT-DTQx** and **PBDTT-DTmFQx** present outstanding thermal stability, with 5% weight loss temperatures (*T*_d_) of 316 and 338 °C under an inert atmosphere, respectively.

### 2.2. Optical Properties

The UV-Vis absorption spectra of **PBDTT-DTQx** and **PBDTT-DTmFQx** in diluted chloroform solutions and thin neat films are presented in Figure 2, and the detailed optical data are listed in Table 2. Both of the copolymers showed two main absorption peaks in CHCl_3_ solution, which is a common feature of D–A type copolymers. The high-lying absorption band from 300 to 500 nm was attributed to the π-π* transition of the conjugated backbone, and the other band from 500 to 800 nm was assigned to the strong intramolecular charge transfer (ICT) state from the donor to the acceptor unit [30]. Meanwhile, both of the copolymers showed similar absorption spectra in CHCl_3_. In comparison with **PBDTT-DTmFQx**, the slight difference is that a shoulder peak centered at ca. 630 nm could be found for **PBDTT-DTQx** in CHCl_3_ solution. It is well known that the fluorine atom has strong electronegativity, but a fluorine substituent shows a weak electron-withdrawing effect due to *meta*-fluorinated phenyl side groups. Compared to the absorption profiles in solution, the ones in thin films showed a certain degree of red-shift, which is ascribed to the intermolecular π-π stacking interaction. Meanwhile, it is noted that the absorption spectra of **PBDTT-DTmFQx**
*vs*
**PBDTT-DTQx** as neat films bathochromically shifted about 15 nm. It is inferred that the interchain interactions among **PBDTT-DTmFQx** chains are stronger than that of **PBDTT-DTQx** chains. The absorption onsets of **PBDTT-DTQx** and **PBDTT-DmFQX** as neat films are located at 730 nm and 800 nm, which are matched with their optical band gaps ($E_{g}^{\mathrm{opt}}$) of 1.73 and 1.68 eV, respectively.

### 2.3. Electrochemical Properties

Highest occupied molecular orbital (HOMO) and lowest unoccupied molecular orbital (LUMO) levels of the donor materials strongly affect the *V*_oc_ and the charge separation efficiency of PSCs. Cyclic voltammetry (CV) was adopted to determine the HOMO and LUMO energy level of the two copolymers, which was determined under nitrogen using *n*-Bu_4_NPF_6_ (0.1 M in anhydrous acetonitrile) as the supporting electrolyte [33]. The ferrocene/ferrocenium (Fc/Fc^+^) redox couple was used as the standard. The CV curves and data are displayed in Figure 3 and Table 2. The HOMO and LUMO energy levels are calculated using the following Equations (1) and (2), where $E_{\mathrm{onset}}^{\mathrm{ox}}$ is the onset oxidation potential and $E_{\mathrm{onset}}^{\mathrm{red}}$ is the onset reduction potential [30].
(1)$$E_{\mathrm{HOMO}}=-\left(4.80-E_{1/2,\mathrm{Fc}/\mathrm{Fc}}+E_{\mathrm{onset}}^{\mathrm{ox}}\right)\;(\mathrm{eV})$$
(2)$$E_{\mathrm{LUMO}}=-\left(4.80-E_{1/2,\mathrm{Fc}/\mathrm{Fc}}+E_{\mathrm{onset}}^{\mathrm{red}}\right)\;(\mathrm{eV})$$

The HOMO energy level of **PBDTT-DTmFQx** (−5.33 eV) is slightly lower than that of **PBDTT-DTQx** (−5.23 eV). Obviously, **PBDTT-DTmFQx**, by grafting two fluorine atoms at phenyl side-chains into the Qx unit, can not only improve the electron-withdrawing ability of Qx, but also cause a lower HOMO level. The comparatively deeper HOMO levels could be expected to achieve higher *V*_oc_ in PSC applications, which is one of main contributors of high efficiency PSCs. The bandgap values ($E_{g}^{\mathrm{ec}}$) in terms of the electrochemical study were 1.74 and 1.70 eV for **PBDTT-DTQx** and **PBDTT-DTmFQx**, respectively, which are consistent with the optical bandgaps.

### 2.4. Photovoltaic Properties of the Copolymers

To investigate the photovoltaic properties of **PBDTT-DTQx** and **PBDTT-DTmFQx**, PSCs were fabricated by blending **PBDTT-DTQx** or **PBDTT-DTmFQx** as the donor and PC_71_BM as the acceptor with the configuration of ITO/PEDOT:PSS/polymer:PC_71_BM/PDINO (perylene diimide functionalized with amino N-oxide)/Al. These PSCs were then tested under AM 1.5G irradiation (100 mW cm^−2^). Photovoltaic performances of the devices were optimized by using different donor/acceptor weight ratios, active layer thicknesses, and solvents with 1,8-diiodooctane (DIO) as the additive. Table 3 lists the *V*_oc_, *J*_sc_, FF, and PCE of the optimized devices. The optimized device fabrication conditions include the donor/acceptor weight ratio of 1:1.5 for **PBDTT-DTQx** and 1:2 for **PBDTT-DTmFQx**, and an active layer thickness of ca. 150 nm. The devices based on **PBDTT-DTQx**:**PC_71_BM** (1:1.5, *w*/*w*) and **PBDTT-DTmFQx:PC_71_BM** (1:2, *w*/*w*) showed *V*_oc_ values of 0.85 V and 0.91 V; *J*_sc_ values of 8.67 mA cm^−2^ and 10.57 mA cm^−2^; and FF values of 33.27% and 39.44%; with PCE values of 2.46% and 3.79%, respectively. The higher *V*_oc_ value of **PBDTT-DTmFQx** compared with **PBDTT-DTQx** was derived from the corresponding lower HOMO value. Compared to **PBDTT-DTQx**, the higher *J*_sc_ and FF values of **PBDTT-DTmFQx** results in superior PCE values. To further optimize the photovoltaic properties of PSCs, DIO was frequently selected as the solvent additive due to its ability to solvate fullerene derivatives as the acceptor and high boiling point. The volume ratio of DIO to DCB varied from 1% to 3% in order to optimize the device performance. We found that the PSCs using 2% DIO as the additive showed the best photovoltaic performances for the polymers. Figure 4 shows current density–voltage (*J*–*V*) curves of the optimal PSCs with or without DIO as the additive. These devices for **PBDTT-DTQx** and **PBDTT-TmFQx** showed *V*_oc_ values of 0.87 and 0.91 V; *J*_sc_ values of 12.00 and 11.05 mA cm^−2^; FF values of 61.45% and 53.95%; and PCE values of 6.40% and 5.43%, respectively. Noticeably, it was found that the addition of DIO as the additive resulted into a slight variation of *V*_oc_ (slight increase for **PBDTT-DTQx** and slight decrease for **PBDTT-DTmFQx**). The variation can be attributed to the change of charge-separated and charge-transfer-state energies derived from morphology evolution of the active layers upon additive addition [34]. Although the device for **PBDTT-TmFQx** with 2% DIO showed a slightly lower *V*_oc_ value, its *J*_sc_ and FF values were higher; this resulted in a higher efficiency owing to improved intermolecular packing.

To further explain the different photovoltaic properties of the copolymers in PSC devices, the hole mobility of the blend films was measured by the space charge limited current (SCLC) model with a typical device structure of ITO/PEDOT:PSS/polymer:PC_71_BM/Au. The SCLC was calculated using the Mott-Gurney law: *J* = (9/8)*ε*_0_*ε*_r_*μ*_h_(*V*^2^/*L*^3^), where *J* stands for current density, *ε*_0_ is the permittivity of free space, *ε*_r_ is the relative dielectric constant of the transport medium, *μ*_h_ is the hole mobility, *V* is the applied voltage (*V*_app_) corrected from the built-in voltage (*V*_bi_) arising from the difference in the work function of the contacts, and *L* is the thickness of the active layer [35]. The *J–V* curves of these hole devices containing copolymers/PC_71_BM active layers are exhibited in Figure 5. As listed in Table 3, the hole mobility of PBDTT-DTQx and PBDTT-DTmFQx were calculated to be 8.38 × 10^−5^ and 2.34 × 10^−4^ cm^2^ V^−1^ s^−1^ in the hole copolymer/PC_71_BM-based devices, respectively. Obviously, the higher hole mobility of PBDTT-DTmFQx:PC_71_BM contributes in part to the higher *J*_sc_ and FF values observed in the PSCs, which could be attributed to enhanced PCE in the devices.

The morphology of the blend films under optimized conditions (2% DIO) could also make the improved photovoltaic performance of the **PBDTT-DTQx** and **PBDTT-DTmFQx**-based device clear. The surface morphology of the copolymers/PC_71_BM blend films was recorded by atomic force microscopy (AFM), and the relevant topographical and phase images in a surface area of 2×2 µm are in presented Figure 6. The root mean square (RMS) roughness from the height images were determined as 2.15 and 1.74 nm for the **PBDTT-DTQx** and **PBDTT-DTmFQx**-based blend films, respectively. Clearly, a smoother surface was observed for the **PBDTT-DTmFQx**-based blend film. Therefore, the higher PCE value for the **PBDTT-DTmFQx**-based device can be ascribed to smoother morphologies, more ordered bicontinuous interpenetrating networks and more efficient charge transport properties (Figure 5), which provided a *J_sc_* of 12.00 mA cm^−2^, a FF of 61.45% and better device performance with a PCE of up to 6.40%.

## 3. Experimental Section

### 3.1. Materials

4,7-Dibromo-2,1,3-benzothidiazole (**1**) was purchased from Suna Tech Inc (Suzhou, Jiangsu, China). The monomer (4,8-bis(5-((2-ethylhexyl)thio)thiophen-2-yl)benzo[1,2-b:4,5-b′]-dithio-phene- 2,6-diyl)bis(trimethylstannane)(**5**) [36], 3,3′-difluorobenzil [37] and tributyl-(3-(2-ethylhexyl)- thiophen-2-yl)stannane [38] were synthesized according to the reported references. The other chemicals and reagents were received from commercial sources and used without further purification.

### 3.2. Characterization

^1^H-NMR and ^13^C-NMR measurements were carried on a Bruker 400 MHz DRX spectrometer (Bruker Analytische Messtechnik GmbH, Rheinstetten, Germany) with tetramethylsilane (TMS) as the internal reference. UV-Vis absorption spectra were recorded on a Cary 60 UV-Vis spectrophotometer (Agilent technologies, Australia). Thermogravimetric analysis (TGA) (TA Instruments Corporate, 159 Lukens Drive New Castle, DE, USA) measurements were performed under nitrogen flow at a heating rate of 20 °C min^−1^. Cyclic voltammetry was carried out on a CHI600A electrochemical workstation (Shanghai, China) using a polymer film on a platinum electrode as the working electrode, a platinum wire as the counter electrode and Ag/AgCl as a reference electrode at a scan rate of 50 mV s^−1^. The surface morphology of the blend film was investigated by AFM on a Veeco, DI multimode NS-3D apparatus (Plainview, NY, USA) in tapping mode under normal air conditions at RT with a 5 mm scanner.

### 3.3. Fabrication and Characterization of PSCs

PSCs were fabricated through the conventional process. The basic device structure was ITO/PEDOT:PSS/polymer:PC_71_BM/PDINO/Al. Patterned ITO glass substrates were sequentially cleaned with detergent, de-ionized (DI) water, acetone, and isopropanol. They were then treated in an oxygen plasma (Plasma Preen II-862 Cleaner, North Brunswick, NJ, USA). A buffer layer of PEDOT:PSS (ca. 40 nm) was spin-coated onto the precleaned ITO substrate and annealed in an oven at 150 °C for 20 min. The photoactive layer was subsequently prepared by spin-coating a solution of the polymer/PC_71_BM in chlorobenzene (CB) with/without DIO additive on the PEDOT:PSS layer with a typical concentration of 10 mg/mL. The thickness of active layers was varied by changing the spin-coating speed, and the optimized thickness was about 120 nm. PDINO was dissolved into methanol at a concentration of 1.5 mg/mL and spin-coated on the top of the photoactive layer. The thickness of PDINO film was about 10 nm. Finally, the Al layer (ca.100 nm) was successively deposited on the PIDNO layer in vacuum and used as the top electrode. The current density–voltage (*J–V*) characterization of the devices was carried out on a computer-controlled Keithley source measurement system. A solar simulator was used as the light source and the light intensity was monitored by a standard Si solar cell. The active area was 0.1 cm^2^ for each cell. The thicknesses of the spuncast films were recorded by a profilometer (Alpha-Step 200, Tencor Instruments, Milpitas, CA, USA).

### 3.4. Synthesis of the Monomers and Copolymers

#### 3.4.1. Synthesis of 5,8-dibromo-2,3-diphenylquinoxaline (**2a**)

In a dry 100 mL flask, compound 1 (0.59 g, 2 mmol) and NaBH_4_ (0.76 g, 20 mmol) were mixed in ethanol (40 mL) for 1 h at RT, and then refluxed for 4 h under a nitrogen atmosphere. After cooling to RT, water was added and filtered. The residue was dried in vacuum. Then the crude product and sodium acetate (0.33 g, 4 mmol) were added to a solution of benzil (0.42 g, 2 mmol) in anhydrous ethanol (30 mL) under a nitrogen atmosphere. The mixture was refluxed for 24 h. After cooling to RT, the solvent was removed under reduced pressure and the residue was purified by silica gel column chromatography (eluent: *n*-hexane:CH_2_Cl_2_ = 4:1) to give a pale yellow solid (yield 72.0%, 0.75 g). ^1^H-NMR (400 MHz, CDCl_3_, δ): 7.91(s, 2H), 7.65–7.66(m, 4H), 7.34–7.41(m, 6H). ^13^C-NMR (100 MHz, CDCl_3_, δ): 154.15, 139.36, 137.95, 133.10, 130.26, 129.59, 128.38, 123.74.

#### 3.4.2. Synthesis of 5,8-dibromo-2,3-bis(3-fluorophenyl)quinoxaline (**2b**)

The compound **2b** was prepared according to the synthetic process of the above compound **2a** and gave a red powder. Yield 78%. ^1^H-NMR (400 MHz, CDCl_3_, δ): 7.95(s, 2H), 7.46(d, *J* = 12 Hz, 2H), 7.26–7.36(m, 4H), 7.11–7.16(m, 2H). ^13^C-NMR (100 MHz, CDCl_3_, δ): 163.71, 161.74, 152.52, 139.68, 139.41, 133.66, 130.04, 126.00. 123.76, 117.03.

#### 3.4.3. Synthesis of 5,8-bis(4-(2-ethylhexyl)thiophen-2-yl)-2,3-diphenylquinoxaline (**3a**)

Under a nitrogen atmosphere, Pd(PPh_3_)_4_ (56 mg) was added to a solution of tributyl(4-(2-ethylhexyl)-thiophen-2-yl)stannane (1.94 g, 4 mmol) and compound **2a** (0.44 g, 1 mmol) in dry toluene (20 mL). The mixture was refluxed for 12 h. After cooling to RT, the solvent was removed under reduced pressure and the residue was purified by silica gel column chromatography (eluent: *n*-hexane: CH_2_Cl_2_ = 5:1) to give a red solid (yield 91%, 0.61 g). ^1^H-NMR (400 MHz, CDCl_3_, *δ*): 8.10(s, 2H), 7.73–7.77(m, 6H), 7.35–7.41(m, 6H), 7.08(s, 2H), 2.58–2.88(m, 4H), 1.65(m, 2H), 1.25–1.42(m, 16H), 0.89–0.93(m, 12H). ^13^C-NMR (100 MHz, CDCl_3_, δ): 151.48, 141.60, 138.77, 138.27, 137.29, 131.20, 130.49, 128.97, 128.65, 128.20, 126.90, 124.66, 40.49, 34.64, 32.59, 29.00, 25.66, 23.11, 14.21, 10.93.

#### 3.4.4. Synthesis of 5,8-bis(4-(2-ethylhexyl)thiophen-2-yl)-2,3-bis(3-fluorophenyl)quinoxaline (**3b**)

The compound 3b was synthesized according to the synthetic process of the above compound **3a** and gave a red powder. Yield 93%. 1H-NMR (400 MHz, CDCl_3_, δ): 8.15(s, 2H), 7.73(s, 2H), 7.58(d, *J* = 12 Hz, 2H), 7.45(d, *J* = 8 Hz, 2H), 7.28–7.38(m, 2H), 7.13–7.16(m, 4H), 2.64(d, *J* = 8 Hz, 4H), 1.67(m, 2H), 1.27–1.43(m, 16H), 0.90–0.95(m, 6H).

#### 3.4.5. Synthesis of 5,8-bis(5-bromo-4-(2-ethylhexyl)thiophen-2-yl)-2,3-diphenylquinoxaline (**4a**)

To a solution of compound **3a** (0.34 g, 0.5 mmol) in CHCl_3_ and acetic acid (10 mL, CHCl_3_: acetic acid =5:1, *v*/*v*) was added NBS (0.20 g, 1.1 mmol) in portions over 20 min. The mixture was stirred at RT in darkness for 12 h and then poured into water, and extracted with CH_2_Cl_2_ for three times. The combined organic layers were dried over anhydrous MgSO_4_. After removing the solvent under reduced pressure, the residue was purified by silica gel column chromatography to obtain a red solid (yield 82%, 0.34 g). ^1^H-NMR (400 MHz, CDCl_3_, δ): 8.06(s, 2H), 7.70–7.72(m, 4H), 7.53(s, 2H), 7.38–7.43(m, 6H), 2.57 (d, *J* = 8 Hz, 4H), 1.70(m, 2H), 1.31–1.40(m, 14H), 0.88–096(m, 12H). ^13^C-NMR (100 MHz, CDCl_3_, δ): 151.95, 140.48, 138.27, 137.45, 136.82, 130.47, 130.26, 129.13, 128.25, 127.13, 125.68, 114.78, 40.07, 33.79, 32.53, 28.84, 25.68, 23.08, 14.16, 10.87.

#### 3.4.6. Synthesis of 5,8-bis(5-bromo-4-(2-ethylhexyl)thiophen-2-yl)-2,3-bis-(3-fluoro-phenyl) quinoxaline (**4b**)

The compound **4b** was synthesized by the same procedure as the compound **4a**. Yield 85%. ^1^H-NMR (400 MHz, CDCl_3_, δ): 8.05(s, 2H), 7.50(s, 2H), 7.36–7.46(m, 6H), 7.12–7.16(m, 2H), 2.56(d, *J* = 8Hz, 4H), 1.68–1.71(m, 2H), 1.31–1.41(m, 16H), 0.88–0.94(m, 12H). ^13^C-NMR (100 MHz, CDCl_3_, δ): 164.32, 161.05, 150.35, 140.70, 140.11, 140.01, 137.16, 136.92, 130.57, 130.04, 129.93, 127.43, 126.24, 117.43, 117.12, 116.58, 116.30, 114.97, 40.10, 33.81, 32.55, 28.85, 25.70, 23.11, 14.19, 10.89.

#### 3.4.7. Synthesis of the Polymer **PBDTT-DTQx**

In a dry 25 mL flask, Pd_2_(dba)_3_, (2.0 mg) and P(*o*-Tol)_3_, (4.0 mg) were added to a solution of the monomer **4a** (0.2 mmol) and **5** (194 mg, 0.2 mmol) in 8 mL degassed toluene under a nitrogen atmosphere, and then stirred vigorously and refluxed for 19 h until the reaction system became a viscous state. After cooling to RT, the mixture was poured into methanol and precipitation occurred. It was collected by filtration and successively extracted in a Soxhlet apparatus with *n*-hexane, acetone and chloroform (CHCl_3_), respectively. The collected CHCl_3_ solution was concentrated and precipitated with methanol to get a dark solid (135 mg, 81.3%). Anal. Calcd for C_78_H_88_N_2_S_8_: C, 71.51; H, 6.77; N, 2.14; S, 19.58. Found: C, 71.45; H, 6.90; N, 2.23; S, 19.34.

#### 3.4.8. Synthesis of the Polymer **PBDTT-DTmFQx**

**PBDTT-DTmFQx** was synthesized by the same procedure as **PBDTT-DTQx**. Yield 89.2%. Anal. Calcd for C_78_H_86_F_2_N_2_S_8_: C, 69.60; H, 6.44; N, 2.08; S, 19.05. Found: C, 70.01; H, 6.32; N, 2.13; S, 19.30.

## 4. Conclusions

In summary, two novel D–A type NBG copolymers of **PBDTT-DTQx** and **PBDTT-DTmFQx** were designed and synthesized. After grafting two F atoms on the Qx unit, the molecular energy level, packing and charge-carrier transport capability of the **PBDTT-DTmFQx** have been finely tuned, which resulted in a higher *V*_oc_, *J*_sc_, FF of the **PBDTT-DTmFQx**-based device compared to that of the nonfluorinated **PBDTT-DTQx**. The *μ*_h_ of **PBDTT-DTmFQx** was 2.8 times more than that of **PBDTT-DTQx**. The fluorinated **PBDTT-DTmFQx**/PC_71_BM-based PSCs presented a maximum PCE of 6.4% with a *V*_oc_ of 0.87 V, *J*_sc_ of 12.0 mA cm^−2^, and FF of 61.45% under AM 1.5G illumination. The results demonstrate that appending the fluorine atom onto the **Qx** unit through side-chain engineering can effectively improve the photovoltaic properties of the corresponding copolymers.

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
