# Peer review of "Steady Enhancement in Photovoltaic Properties of Fluorine Functionalized Quinoxaline-Based Narrow Bandgap Polymer"

_molecules, 2018, doi:10.3390/molecules24010054_

Reviewer 1 Report

In this work, two copolymers were synthesized and employed as the donors in OPV devices. Compared with the devices based on non-fluorine substituted PBDTT-DTQx, the devices based on fluoride PBDTT-DTmFQx exhibited better performance. Overall, the novelty and importance of this work is high, which warranty the publishing on Molecules. Some comments are shown as below:

1.      To help readers to study the background, two recent review papers of OPV field are recommended to cite (Nat. Photonics, 2018, 12, 131; Chem. Soc. Rev., 2016, 45, 2544).

2.      The film absorption edges of two new donors are around 700 nm, which can be good match to non-fullerene acceptors (such as ITIC-4F). Have the authors compared the performance of devices based on fullerene and non-fullerene acceptors?

3.      After the addition of 2% DIO, the VOC of PBDTT-DTQx based device increased. However, the VOC of PBDTT-DTmFQx based device decreased. This phenomenon may link to film morphology and/or charge transfer state. Please add more discussion.

Author Response

Comment :1.To help readers to study the background, two recent review papers of OPV field are recommended to cite (Nat. Photonics, 2018, 12, 131; Chem. Soc. Rev., 2016, 45, 2544).

Response: We are thankful to the meaningful and constructive comment of referee. These literatures have been citied in introduction part of main-text and highlighted in red.

Comment 2.The film absorption edges of two new donors are around 700 nm, which can be good match to non-fullerene acceptors (such as ITIC-4F). Have the authors compared the performance of devices based on fullerene and non-fullerene acceptors?                                                     

Response. We are regret that the performance of devices based non-fullerene acceptors have not been tried, and we will continue it in the future. We are thankful to the meaningful and constructive comment of referee.

Comment 3. After the addition of 2% DIO, the VOC of PBDTT-DTQx based device increased. However, the VOC of PBDTT-DTmFQx based device decreased. This phenomenon may link to film morphology and/or charge transfer state. Please add more discussion.

Response: We are thankful to the meaningful and constructive comment of referee. We have added more discussion, please see the “Results and Discussion” part of main-text and highlighted in red.

Reviewer 2 Report

In the manuscript the authors describe two novel D-A type NBG copolymers synthesized and characterized from 2-ethylhexylthiothiophene-substituted benzodithiophene, 2,3-diphenylquinoxaline or 2,3-bis(3-fluorophenyl)quinoxaline and 2-ethylhexylthiophene. By modification of Qx system molecular energy level, packing and charge-carrier transport capability resulted tuned. In particular, the fluorine atom in side-chain effectively caused an improvement of the photovoltaic properties of the corresponding copolymers. 

The manuscript contains wide and detailed information about synthesis and characterization of the new polymeric system and appropriate description of the fabrication of PSCs. Electrochemical and optical properties were minutely examined, including HOMO-LUMO calculations and photovoltaic properties. AFM images also provide relevant morphological information. 

In Pf.1, line 77-79, the authors give photovoltaic data of PBDTT-DTmFQx/PC71BM-based device which presents a maximum PCE of 6.40% obtained with a Jsc of 12.0 mA cm-2 The optimization of performances in inverted cells respect to conventional PSCs is actually a goal for researches. It could be interesting to ensure the completion of the work to test the system at increased Jsc in inverted devices. About that it might be useful to add literature on the study of improving output current in inverted PSCs such as: “Elucidating the origin of the improved current output in inverted polymer solar cells”, by P.Morvillo, R. Ricciardi, G. Nenna, E. Bobeico, R. Diana, C. Minarini- Solar Energy Materials & Solar Cells, 152 (2016) 51–58.

The work actually provides a valid contribution in the field of modified polymers based PSCs. Characterization and discussion appears exhaustive and congruent. The manuscript is well written in proper English (but too much commas). At this stage the paper can be considered for publication on Molecules.

Author Response

Comment .Pf.1, line 77-79, the authors give photovoltaic data of PBDTT-DTmFQx/PC71BM-based device which presents a maximum PCE of 6.40% obtained with a Jsc of 12.0 mA cm-2 The optimization of performances in inverted cells respect to conventional PSCs is actually a goal for researches. It could be interesting to ensure the completion of the work to test the system at increased Jsc in inverted devices. About that it might be useful to add literature on the study of improving output current in inverted PSCs such as: “Elucidating the origin of the improved current output in inverted polymer solar cells”, by P.Morvillo, R. Ricciardi, G. Nenna, E. Bobeico, R. Diana, C. Minarini- Solar Energy Materials & Solar Cells, 152 (2016) 51–58.

Response: We are thankful to the meaningful and constructive comment of referee. These literature has been citied in “Results and Discussion” part of main-text and highlighted in red.